# Aptamer Targets Triple-Negative Breast Cancer through Specific Binding to Surface CD49c

**DOI:** 10.3390/cancers14061570

**Published:** 2022-03-18

**Authors:** Quanyuan Wan, Zihua Zeng, Jianjun Qi, Yingxin Zhao, Xiaohui Liu, Zhenghu Chen, Haijun Zhou, Youli Zu

**Affiliations:** 1Department of Pathology and Genomic Medicine, Houston Methodist Hospital, Houston, TX 77030, USA; qyuanwan@gmail.com (Q.W.); zzeng@houstonmethodist.org (Z.Z.); jqi@houstonmethodist.org (J.Q.); lcswyh@126.com (X.L.); zchen2@houstonmethodist.org (Z.C.); hzhou@houstonmethodist.org (H.Z.); 2Department of Internal Medicine, University of Texas Medical Branch, Galveston, TX 77555, USA; yizhao@utmb.edu

**Keywords:** aptamer ligand, biomarker identification, integrin α3/CD49c, triple-negative breast cancer (TNBC), targeted cancer therapy

## Abstract

**Simple Summary:**

Targeted therapy directed against many biomarkers has not shown significant improvement in outcome in TNBC, and therefore it is urgent to discover more biomarker candidates. Here, we found a DNA aptamer that bound to TNBC cells and identified CD49c as a specific surface marker for TNBC cells using the aptamer-facilitated biomarker discovery technology. The findings suggest that this DNA aptamer can be a drug delivery vehicle and CD49c is a potential target of targeted therapy for TNBC.

**Abstract:**

Although targeted cancer therapy can induce higher therapeutic efficacy and cause fewer side effects in patients, the lack of targetable biomarkers on triple-negative breast cancer (TNBC) cells limits the development of targeted therapies by antibody technology. Therefore, we investigated an alternative approach to target TNBC by using the PDGC21T aptamer, which selectively binds to poorly differentiated carcinoma cells and tumor tissues, although the cellular target is still unknown. We found that synthetic aptamer probes specifically bound cultured TNBC cells in vitro and selectively targeted TNBC xenografts in vivo. Subsequently, to identify the target molecule on TNBC cells, we performed aptamer-mediated immunoprecipitation in lysed cell membranes followed by liquid chromatography tandem mass spectrometry (LC-MS/MS). Sequencing analysis revealed a highly conserved peptide sequence consistent with the cell surface protein CD49c (integrin α3). For target validation, we stained cultured TNBC and non-TNBC cells with an aptamer probe or a CD49c antibody and found similar cell staining patterns. Finally, competition cell-binding assays using both aptamer and anti-CD49c antibody revealed that CD49c is the biomarker targeted by the PDGC21T aptamer on TNBC cells. Our findings provide a molecular foundation for the development of targeted TNBC therapy using the PDGC21T aptamer as a targeting ligand.

## 1. Introduction

Breast cancer is the most common cancer among women worldwide, and is the fifth leading cause of cancer mortality, with an estimated 2.3 million new cases representing 11.7% of all cancer cases in 2020 [1]. Breast cancers are categorized into six subtypes based on the expression of cell surface molecular markers. For therapeutic purposes, breast cancers are categorized into three subtypes: estrogen and progesterone receptor (ER, PR)-positive, human epidermal growth factor receptor 2 (HER2)-positive, and triple (ER, PR, and HER2)-negative [2]. The triple-negative breast cancer (TNBC) subtype accounts for 15–20% of all breast cancers. Clinically, TNBC grows and spreads faster than other breast cancer subtypes and therefore tends to be more aggressive, with a higher potential for disease metastasis and recurrence. Pathologically, TNBC tumor cells are poorly differentiated and do not express ER, PR, or HER2. Therefore, TNBC does not respond to ER/PR-based hormonal therapy or HER2-targeting therapies, which are used clinically to treat other types of breast cancers. The lack of specific biomarkers on TNBC cells impedes the development of targeted therapeutic approaches. Currently, a combination of surgery and chemotherapy is the standard of care to treat TNBC. However, none of these are specific for TNBC, and chemotherapy may have severe off-target toxicity in normal tissues [2,3].

Nucleic acid aptamers are small-molecule ligands comprising short, single-strand DNA (ssDNA) or ssRNA. Target-specific aptamers can be developed from ssRNA/ssDNA libraries via a defined experimental process called the systematic evolution of ligands by exponential enrichment (SELEX). Aptamers can be developed via the SELEX approach by using cells as the targets, namely cell-based SELEX (cSELEX). Importantly, the cSELEX allows researchers to develop aptamers specific for cells of interest without knowledge of target molecules on cell surface or no known targetable biomarkers available [4,5,6]. Such aptamers developed by cSELEX have been studied for in vivo tumor imaging, including B-cell lymphoma [7], colorectal cancer [8], and lung cancer [9]. In addition, the aptamers can function as a vehicle for targeted delivery of miRNA/siRNA or anticarcinogens [10], and act as an anti-cancer agent [11,12]. Moreover, the aptamers developed by the cSELEX approach are valuable tools to identify new specific biomarkers on cells of interest via the aptamer-facilitated biomarker discovery (AptaBiD) technology [13]. A recent study using the cSELEX approach identified the ssDNA aptamer (PDGC21T), which recognizes and selectively binds to poorly differentiated gastric cancer tissue in patient specimens, even though the identity of the target molecule(s) was unknown [14]. Because TNBC is also poorly differentiated, we explored whether the aptamer could target TNBC cells. Both in vitro and in vivo studies demonstrated that the synthetic PDGC21T aptamer could bind TNBC cells with high affinity and target TNBC cell-xenograft tumors. To identify the target of the PDGC21T aptamer, we conducted an aptamer-mediated co-precipitation study and subsequent mass spectrometry-based proteomics assays. Peptide identification analysis revealed that the target of PDGC21T is CD49c (integrin alpha 3) on TNBC cells. These findings provide a foundation for the development of an aptamer-guided approach for imaging detection and targeted therapy in TNBC tumors.

## 2. Materials and Methods

### 2.1. Cell Culture

The TNBC cell lines were kind gifts from Dr. Jenny C. Chang’s lab at the Houston Methodist Academic Institute. The other cells were purchased from American Type Culture Collection (ATCC, Manassas, VI, USA), cultured, and stored in our laboratory. Cell lines were cultured in Roswell Park Memorial Institute (RPMI)-1640 medium (Corning, Corning, NY, USA) supplemented with 10% FBS (Corning), 100 units/mL penicillin (Gibco, Waltham, MA, USA), and 100 μg/mL streptomycin (Gibco). For cell passage, 0.25% trypsin with ethylenediaminetetraacetic acid (Corning) and non-enzymatic cell dissociation solution (Corning) were used. All cells were grown at 37 °C in 95% air/5% CO_2_. All experiments were performed with mycoplasma-free cells as tested with e-Myco™ plus mycoplasma PCR detection kit (iNtRON Biotechnology, Seoul, South Korea).

### 2.2. Aptamer Binding Assay

All aptamers, including 5’-dye-labeled or 5’-biotinylated aptamers, were purchased from Integrated DNA Technologies (IDT Inc., Coralville, IA, USA). Dry DNA oligos were dissolved to a storage concentration of 100 μM with PM buffer (1 × PBS (Corning) supplemented with 5 mM MgCl_2_ (Sigma-Aldrich, St. Louis, MO, USA)) and stored at −20 °C until further use. Before use, aptamers were thawed and diluted to 10 μM with PM buffer, heated to 95 °C, and immediately cooled in ice for at least 10 min. Before cell-binding assays, the biotinylated aptamer used for the binding test was incubated with streptavidin-conjugated Cy3 (Invitrogen, Carlsbad, CA, USA) at 37 °C for 30 min to link the aptamer with the Cy3 dye.

Cells were harvested using non-enzymatic cell dissociation solution and then washed once with PBS. Then, cells were distributed into Falcon^®^ 5 mL round-bottomed polystyrene test tubes (Corning) and incubated with varying concentrations of dye-labeled aptamers, random ssDNA sequences, antibodies, or IgG control in 100 μL of binding buffer (1 × PBS supplemented with 0.1 g/L yeast tRNA (Invitrogen, Waltham, MA, USA), 4.5 g/L glucose (Gibco), 1.0 g/L bovine serum albumin (Thermo Scientific, Waltham, MA, USA), and 5 mM MgCl_2_ (Sigma-Aldrich)). After incubation at room temperature (RT) for 15–30 min, cells were collected by means of centrifugation at 400× *g* for 5 min. Then, cells were resuspended with 1 mL of washing buffer (1 × PBS supplemented with 4.5 g/L glucose and 5 mM MgCl_2_) and washed once. Subsequently, cells were collected by means of centrifugation and re-suspended in flow cytometer running buffer (1 × PBS supplemented with 2% FBS and 5 mM MgCl_2_). Flow cytometry was then performed using a BD LSR II Flow Cytometer (BD Biosciences, San Jose, CA, USA) to measure the fluorescence signal from cell-bound aptamers or antibodies. The fluorescence was observed through fluorescent microscopic imaging (Olympus IX81, Olympus America, Melville, NY, USA). Flow cytometry data were analyzed using FlowJo software (version: X 10.0.7r2, FlowJo, Ashland, OR, USA).

### 2.3. Aptamer-Antibody Competition Binding Assay

To determine whether PDGC21T affects anti-CD49c binding, a final concentration of 1 μM Cy3-labeled PDGC21T or a random ssDNA library (negative control) was pre-incubated with MDA-MB-231 or HCC38 cells at RT for 15 min. Then, 1 μL of PE-labeled anti-CD49c (Clone C3 II.1, BD Pharmingen™, San Diego, CA, USA) was added to cells and incubation continued for 15 min. As controls, PE-labeled anti-CD49c or IgG isotype (κ Isotype control, Clone MOPC-21, BD Pharmingen™) were incubated with MDA-MB-231 or HCC38 cells at RT for 15 min. To determine whether anti-CD49c affects PDGC21T binding, 5 μL of PE-labeled anti-CD49c or IgG isotype were pre-incubated with MDA-MB-231 or HCC38 cells at RT for 15 min. Then, fluorescein amidite (FAM)-labeled PDGC21T was added to cells at a final concentration of 200 nM and incubation continued for 15 min. As a blank control, FAM-labeled PDGC21T or a random library was incubated with MDA-MB-231 or HCC38 cells at RT for 15 min. After incubation, cells were washed once with wash buffer and re-suspended in running buffer for flow cytometry.

### 2.4. PDGC21T Pull-Down Assay

Aptamer-based pull-down and high-performance liquid chromatography mass spectrum (HPLC-MS) were used to identify PDGC21T targets. The pull-down assay was performed according to a previously published procedure using a biotinylated aptamer [15]. To measure the binding ability of the biotinylated PDGC21T, two approaches were used. One approach is to link the biotinylated PDGC21T with the streptavidin-Cy3, as described previously. Another approach is to perform the competitive assay. Briefly, a final concentration of 1 μM biotinylated PDGC21T was pre-incubated with MDA-MB-231 cells at RT for 15 min. Then, Cy3-labeled PDGC21T was added to cells at a final concentration of 200 nM and incubation continued for 15 min. Binding ability was measured using flow cytometry. Pull-down was performed after confirming the binding ability of the biotinylated PDGC21T.

Ten million cells were harvested using trypsin and washed once with PBS. After washing, cells were re-suspended and lysed in 1.8 mL of 50 mM Tris-Cl buffer (pH 7.4) (Sigma-Aldrich) supplemented with phenylmethylsulfonyl fluoride (PMSF) (Sigma-Aldrich) and proteinase inhibitor cocktail (Sigma-Aldrich) for 30 min at 4 °C. After incubation, cell debris was collected by means of centrifugation at 12,000× *g* for 3 min at 4 °C and washed three times with 50 mM Tris-Cl buffer (pH 7.4) supplemented with PMSF. The pellet was resuspended and further lysed with 600 μL of lysis buffer (1 × PBS, supplemented with 5 mM MgCl_2_, 1% Triton X-100 (Sigma-Aldrich), PMSF, and proteinase inhibitor cocktail) for 30 min at 4 °C. After lysing, the supernatant was collected by means of centrifugation at 12,000× *g* for 5 min at 4 °C. The supernatant was aliquoted into three clean 2 mL tubes and pre-heated biotin-labeled PDGC21T aptamer or a random ssDNA library was added (final aptamer concentration, 2 μM). Rotated incubation was performed at 4 °C overnight. The next day, pre-washed streptavidin-sepharose (GE Healthcare, Marlborough, MA, USA) (200 μL of original volume/tube) was added and tubes were incubated with rotation for 1 h at 4 °C. After incubation, agarose beads were collected and washed three times with lysis buffer by centrifugation at 12,000× *g* for 5 min at 4 °C. Finally, 10 μL of 5× SDS sample loading buffer was added into each tube containing the collected agarose beads, and samples were heated at 85 °C for 10 min. Supernatant was collected by centrifugation at 12,000× *g* for 5 min at RT. A 15 μL aliquot of supernatant was loaded into the corresponding wells of a 4–15% Mini-PROTEAN^®^ TGX™ precast protein gel (Bio-Rad, Hercules, CA, USA) and electrophoresed at 90 V for 2 h. The silver stain procedure for the gel was conducted according to the manufacturer’s instructions (Pierce^TM^ Silver Stain kit; Thermo Scientific). Protein bands of interest were excised for further liquid chromatography tandem mass spectrometry (LC-MS/MS) analysis (Appendix A) [16,17,18,19].

### 2.5. PDGC21T PEGylation

PDGC21T PEGylation was performed as described previously with minor modifications [20]. Briefly, 120 µg of ^IRD800CW^PDGC21T was dissolved in 800 µL of 0.1 M NaHCO_3_/CH_3_CN (v/v = 1:1, pH 9.0) (Sigma-Aldrich), and 20 mg of mPEG-NHS (Nanocs, New York, NY, USA) was added to this solution. The mixture was incubated overnight at RT with gentle shaking. The reaction was monitored by reversed-phase HPLC (RP-HPLC) using a PRP-1 analytic column (4.1 mm × 150 mm, 10 µm, HAMILTON Corporation, Reno, NV, USA) at a flow rate of 1 mL/min. A 15 min linear gradient from 0% solvent A [100 mM triethylammonium acetate (Sigma-Aldrich) in water, pH 7] to 100% solvent B [5% solvent A in CH_3_CN (Sigma-Aldrich)] was used for each HPLC run. The resulting ^IRD800CW^PDGC21T_PEG5000_ was purified using a PRP-1 semi-preparative column (10 mm × 250 mm, 10 µm, HAMILTON Corporation) at a flow rate of 3 mL/min. An 18 min linear gradient from 0% solvent A (100 mM triethylammonium acetate in water, pH 7) to 85% solvent B was used for each HPLC run. The fractions of ^IRD800CW^PDGC21T_PEG5000_ were collected and solvents were removed with SpeedVac concentrator (SPD131DDA, Thermo Scientific). The purified ^IRD800CW^PDGC21T_PEG5000_ was confirmed with a sodium dodecyl sulphate-polyacrylamide gel electrophoresis (SDS-PAGE) assay.

### 2.6. Tumor Xenograft and In Vivo and Ex Vivo PDGC21T Targeting

For PDGC21T targeting, NOD.Cg-*Prkdc^scid^ Il2rg^tm1Wjl^*/SzJ (NSG) mice from the Jackson Laboratory were subcutaneously injected with 3 × 10^6^ of either HCC1937, SUM159PT, MDA-MB-231, HCC38, Hs578T, or MCF7 breast cancer cells in the lower back. Tumor size was measured using a vernier caliper and tumor volume was calculated using the following formula: (1)$$\mathrm{tumor}\;\mathrm{volume}=\;\mathrm{tumor}\;\mathrm{length}\;\times {\mathrm{tumor}\;\mathrm{width}}^{2}\times 0.52$$

Two months after tumor inoculation or when tumor volume reached 0.5 cm^3^, tumor-bearing mice were injected intravenously with a 0.067 nmol/dose (1.33 μg/dose) of ^IRD800CW^PDGC21T or ^IRD800CW^PDGC21T_PEG5000_. Before injection, aptamers were heated at 95 °C for 5 min followed by an ice bath for at least 10 min. In vivo imaging was performed using an in vivo imaging system (Xenogen IVIS-200, Caliper Life Sciences, Hopkinton, MA, USA) at 30 min, 4 h, and 24 h post-aptamer injection. IRDye800CW-conjugated aptamers were observed under fluorescent settings at excitation and emission wavelengths of 745 and 820 nm, respectively. After in vivo imaging at 24 h post-aptamer injection, mice were euthanized. The heart, lung, kidney, liver, spleen, and tumor were removed and imaged. All images were analyzed using Living Image Software (version 4.7.4, Caliper Life Sciences, Waltham, MA, USA).

### 2.7. Xenograft Tumor Cells Isolation

Tumors were mechanically fragmented by cutting with sterile scissors in a DMEM cell culture medium. The cell-containing DMEM medium was filtered through 40 μm cell strainers. After harvesting by means of centrifugation at 300× *g* for 3 min, cells were incubated in the non-enzymatic cell dissociation solution (Corning) at 37 °C for 10 min. Then, cells were harvested and re-suspended in the binding buffer for the aptamer or antibody binding assay as described above.

## 3. Results

### 3.1. Synthetic PDGC21T Aptamer Specifically Binds to TNBC Cells

The PDGC21T aptamer was synthesized and labeled with fluorescent reporters: 5′-ACACCAAAATCGTCCGTTTCGTTTTAGTCCGTCTCTTTAGGGTGT-3′ [14]. For cell binding assays, cultured TNBC cells, including HCC1937, MDA-MB-231, HCC38, and Hs578T cell lines, and non-TNBC cell lines, including T47D and MCF7, were treated with synthetic aptamers for 30 min at RT. Resultant binding of FAM-labeled and Cy3-labeled aptamers to suspended and adherent tumor cells was examined by fluorescent microscopy and flow cytometry, respectively. The PDGC21T aptamer selectively bound to suspended and adherent TNBC cells but did not react with non-TNBC cells under the same conditions (Figure 1). No cell binding was observed in control experiments with the same length of ssDNA probe containing random sequences.

### 3.2. PDGC21T Aptamer Targets Xenograft TNBC Tumors

In an in vivo targeting study, mouse models with xenograft tumors derived from TNBC cells were developed. Aptamers were labeled with the near-infrared dye IRD800CW at the 5′-end to formulate ^IRD800CW^PDGC21T. Because the polyethylene glycol (PEG) modification can increase the stability and prolong the blood circulation half-life of oligonucleotides [21], ^IRD800CW^PDGC21T aptamers were conjugated with 5 kDa PEG at the 3′-end (Figure 2A), and resultant ^IRD800CW^PDGC21T_PEG5000_ aptamers were purified using RP-HPLC (Appendix A). ^IRD800CW^PDGC21T_PEG5000_ aptamer cell binding was initially tested in vitro as earlier described. Flow cytometry analysis revealed that ^IRD800CW^PDGC21T_PEG5000_ aptamers bound TNBC cells (MDA-MB-231 and HCC38) but did not react with MCF7 cells (Figure 2B).

In an in vivo targeting study, mouse models of TNBC were developed using HCC1937, HCC38, MDA-MB-231, SUM159PT, and Hs578T xenograft tumors. In a control group, xenograft tumors derived from non-TNBC cells (MCF7) were used. Once xenograft tumors reached ≥0.5 cm^3^, ^IRD800CW^PDGC21T_PEG5000_ aptamers were administered via the tail vein (Figure 3A). In an in vivo biostability study, ^IRD800CW^PDGC21T aptamers were analyzed. Mice underwent optical imaging scans at 30 min, 4 h, and 24 h post-aptamer administration. Both aptamer probes specifically targeted MDA-MB-231 tumors with detectable imaging signals (Figure 3B). Notably, relative to ^IRD800CW^PDGC21T aptamers, ^IRD800CW^PDGC21T_PEG5000_ aptamers showed superior enhancement in MDA-MB-231 tumors and imaging signals lasted longer, up to 24 h post-aptamer administration. Specific TNBC tumor targeting by ^IRD800CW^PDGC21T_PEG5000_ aptamers was also confirmed in mice bearing xenograft tumors derived from HCC1937, HCC38, SUM159PT, or Hs578T cells (Appendix A). To rule out non-specific imaging, ^IRD800CW^PDGC21T_PEG5000_ aptamers were administered to mice with non-TNBC xenograft tumors derived from MCF7 cells, and a whole-body imaging scan was performed as earlier described. Though weak signals were detected peripherally in MCF7 tumor sides at 30 min post-aptamer administration, they faded rapidly and became undetectable (Figure 3C).

To confirm the in vivo imaging results, xenograft tumors and major organs were collected after whole-body imaging. Ex vivo imaging demonstrated that mice treated with ^IRD800CW^PDGC21T_PEG5000_ aptamers showed stronger signal enhancement in TNBC tumors relative to those treated with ^IRD800CW^PDGC21T aptamers (Figure 3D,E, and Appendix A). In contrast, no signals were detected in non-TNBC tumors in mice treated with ^IRD800CW^PDGC21T_PEG5000_ aptamers (Figure 3F), though a similar pattern of the background signal was present. Taken together, these findings indicate that PDGC21T aptamers selectively target TNBC xenograft tumors and PEGylation significantly improves the circulation half-life of PDGC21T aptamers.

### 3.3. PDGC21T Targets Human CD49c

Because the PDGC21T aptamer can selectively bind intact TNBC cells and xenograft tumors, its target is considered a cell surface molecule or membrane protein. To discover the identity of the PDGC21T aptamer target on tumor cells, biotinylated PDGC21T was synthesized. First, the cell-binding capacity of the biotinylated aptamer was confirmed in MDA-MB-231 cells using a Cy3-labeled aptamer probe as a control (Figure 4A). Subsequently, the membrane protein fraction derived from TNBC cells (MDA-MB-231, HCC38, and Hs578T mixtures) was prepared, and target co-precipitation by biotinylated aptamer was performed. As a background control, biotinylated ssDNA containing a random sequence of the same length as the PDGC21T aptamer was used. The resultant co-precipitated complexes were pulled down with streptavidin-conjugated agarose beads, separated by SDS-PAGE, and developed with silver staining. A protein band with a relative molecular weight of ~118 kDa was detected in a co-precipitated complex of PDGC21T aptamer, but not the random ssDNA control or blank sample (Figure 4B and Appendix A). The same set of co-precipitation assays were carried out using membrane fractions of non-TNBC cells (mixture of MCF7, T47D, and Jurkat cells), and this protein band was not detected in samples incubated with PDGC21T aptamer or the random ssDNA control (Figure 4B).

The protein band was collected for label-free LC-MS/MS analysis. A total of 64 proteins with <1% false-positive rate (FDR) were detected and quantified (Appendix A). Student’s *t*-test with permutation-based FDR correction was used to identify proteins whose abundance varied significantly between the two samples (PDGC21T aptamer vs. random ssDNA). Integrin α3 (CD49c, encoded by *ITGA3*) was the only protein significantly (fold change = 1143, *p* = 1.3 × 10^−18^) enriched in the PDGC21T aptamer co-precipitation complex, indicating a specific interaction between the PDGC21T aptamer and CD49c (Figure 4C). Six unique CD49c peptides were identified in this experiment. The annotated MS/MS spectra of one of the identified CD49c peptides with the sequence STEVLTCATGR are shown in Figure 4D.

To confirm that PDGC21T aptamer targeted TNBC cells through cell surface CD49c, a CD49c antibody (anti-CD49c) was used as a cell-binding control. Ten TNBC and eight non-TNBC cell lines were treated with the PDGC21T aptamer or anti-CD49c under identical conditions, and resultant cell staining patterns were compared. Flow cytometry analysis revealed that PDGC21T selectively targeted TNBC cells with high binding affinity. A nearly identical staining pattern to that obtained with anti-CD49c was observed (Figure 5A). In contrast, both the aptamer and antibody had little/no binding to non-TNBC cells, which express almost no CD49c (Figure 5B). The resultant mean fluorescent intensity of cell binding was quantified and graphed for the comparison of both the aptamer and antibody (Figure 5C,D, and Appendix A).

Finally, to determine whether the aptamer and antibody were targeting the same site on CD49c, competition binding assays were performed. To evaluate whether anti-CD49c competes with PDGC21T aptamer for cell binding, TNBC cells were pre-incubated with anti-CD49c or IgG isotype control and treated with FAM-labeled PDGC21T aptamer. Flow cytometry analysis showed that PDGC21T aptamer binding to MDA-MB-231 (Figure 6A) and HCC38 cells (Figure 6B) was not affected by pre-incubation with anti-CD49c. In contrast, when TNBC cells were pre-incubated with PDGC21T aptamer, PE-labeled anti-CD49c binding to MDA-MB-231 (Figure 6C) and HCC38 cells (Figure 6D) was significantly reduced, indicating that the binding of PDGC21T aptamers to cells impedes the interaction of the anti-CD49c antibody with cell-surface CD49c, indicating that the PDGC21T aptamer targets CD49c.

To further confirm that the targeting of PDGC21T to TNBC cell-xenograft tumors was associated with the expression of CD49c, these xenograft tumors were collected and dissociated into single-cell suspensions, followed by being incubated with the Cy3-labeled PDGC21T or the PE-labeled anti-CD49c antibody (Appendix A). The binding of the aptamer or the antibody was measured using flow cytometry. The results reveal that both the PDGC21T and the anti-CD49c antibody could bind to MDA-MB-231 xenograft-tumor cells (Appendix A) but not MCF7 xenograft-tumor cells (Appendix A), suggesting that the PDGC21T aptamer can target those TNBC-xenograft tumor cells with high expression of CD49c.

## 4. Discussion

Targeted anti-tumor agent delivery is a promising precision therapy for cancer treatment, as it allows the accumulation of anti-tumor agent in tumors via the enhanced permeability and retention effect and active cellular uptake, which reduces adverse side effects while improving therapeutic efficacy [22]. A targeted anti-tumor agent delivery system is principally composed of a targeting ligand that binds a specific surface marker on cancer cells, a spacer and linker, and an optimal anti-tumor agent [23]. Targeting ligands can be nucleic acid aptamers or antibodies. For tumors with known specific cell surface markers, antibodies can serve as excellent targeting ligands [24]. Several clinical trials are investigating antibody-drug conjugates for TNBC therapy [25]. However, for tumors lacking targetable cell surface markers, nucleic acid aptamers may act as ligands for targeted cancer therapy [26]. Indeed, cell-specific aptamers can be readily developed using the cell-SELEX procedure [27].

To detect poorly differentiated gastric cancer cells lacking a targetable molecular marker, Li et al. used the cell-SELEX procedure to develop a 45-nt DNA aptamer (PDGC21T) that selectively targeted poorly differentiated gastric cancer cells [14]. We demonstrated that this aptamer could also target TNBC cell lines in both PBS-based binding buffer and complete DMEM cell culture medium, which suggested that the structure of PDGC21T aptamer was stable in physiological environments. Next, we tested whether the PDGC21T aptamer could target TNBC tumor xenografts in mice. Because PDGC21T binding to MDA-MB-231 cells was stronger relative to other cells (Figure 1), MDA-MB-231 tumor xenografts were used to test the targeting ability of IRDye800CW-labeled PDGC21T [28]. PDGC21T targeted MDA-MB-231 tumor xenografts, but dramatic signal attenuation occurred within 24 h, indicating a short retention time in vivo. As unmodified aptamers are susceptible to nuclease-mediated degradation [29], PDGC21T requires chemical modification to extend the circulation time. PEGylation is an eminent aptamer modification used clinically and in research [21,30]. For therapeutic study, PEGylated PDGC21T aptamer was produced [20], and a validation study revealed that the PEGylated PDGC21T aptamer specifically bound to TNBC cells (MDA-MB-231 and HCC38) and did not react with off-target non-TNBC cells (MCF7). In vivo targeting tests revealed that retention of PEGylated PDGC21T in the liver and spleen exceeded that of non-PEGylated PDGC21T. Furthermore, tumors accumulated more PEGylated PDGC21T and retained the modified aptamers for longer durations relative to non-PEGylated PDGC21T. Further in vivo and ex vivo targeting tests revealed that PEGylated PDGC21T targeted and remained in TNBC xenograft tumors more efficiently than in non-TNBC xenograft tumors. Together, these findings indicate that PEGylated PDGC21T is a promising targeting ligand to facilitate the delivery of anti-TNBC tumor agents.

In addition to their roles as targeting ligands, aptamers are effective agents for cancer biomarker discovery [31,32]. We found that CD49c was the exclusive molecular target of PDGC21T. Cell-binding assays confirmed that PDGC21T exhibited cell-binding patterns mirroring those obtained with a CD49c-specific antibody. A competition assay revealed that while PDGC21T hindered anti-CD49c binding, anti-CD49c did not affect PDGC21T binding, indicating that the PDGC21T-binding region on CD49c overlaps with the antibody-binding region of CD49c. These data verified that CD49c is the target of PDGC21T. CD49c was previously identified as a highly expressed cell surface protein in some TNBC cell lines, including MDA-MB-231, MDA-MB-436, MDA-MB-157, HCC1143, HCC1937, and SUM149PT [33,34]. Furthermore, *ITAG3*, which encodes CD49c, was characterized as a highly expressed gene in pancreatic cancer patients [35]. We found that CD49c was expressed to a lesser degree on the surface of AsPC-1 and MDA-Panc-28 pancreatic adenocarcinoma cell lines (Figure 5B). Although we identified the high expression of CD49c on the surface of TNBC cell lines, CD49c expression in clinical TNBC samples should be further examined.

CD49c is also known as integrin α3 and interacts with integrin β1 (CD29) to form the VLA3 integrin receptor that regulates cell adhesion [36,37]. CD49c is considered as a strong contributing factor to tumor invasion. Robust expression levels of VLA3 are related to highly migratory and invasive phenotypes in melanoma [38], head and neck cancer [39], pancreatic cancer [40], intrahepatic cholangiocarcinoma [41], TNBC [42], glioma stem-like cells [43], skin tumor formation, and TNBC and oral squamous cell carcinoma metastases [42,44,45]. Clinical analysis revealed that abundant CD49c was associated with poor prognosis in patients with non-small cell lung cancer [46]. CD49c was also identified as a biomarker of cells undergoing an epithelial-mesenchymal transition in breast cancer [47,48]. Because CD49c promotes tumor progression, reducing CD49c levels using targeted siRNA delivery or blocking CD49c using antibodies or aptamers is a feasible approach to prevent cancer metastasis and progression. In a preclinical study, a monoclonal antibody against aberrantly glycosylated integrin α3β1 was used to block integrin signaling transduction, leading to bladder cancer cell-cycle arrest [49]. Another study used the antibody against activated laminin, the ligand of integrin α3β1, to block the activation of integrin α3β1 signaling to prevent dormant cancer cell awakening [50]. The study suggested that CD49c blockade may help to prevent breast cancer cell awakening.

During PDGC21T target identification in this study, we found that PDGC21T hinders antibody binding to CD49c, but antibody pre-treatment did not influence the binding of PDGC21T aptamer to CD49c. These findings suggest two possible interaction mechanisms of aptamer and antibody to bind CD49c. One possibility is that the aptamer may have multiple binding sites on CD49c protein and one of the sites is shared with the CD49c antibody. Another possibility is that because integrins are well known as conformation changing molecules when interacting with their ligands [51,52,53], the aptamer binding may trigger a change in the conformation of CD49c protein and thus may have an adverse impact on antibody binding. To testify the first model, CD49c proteins should be gradually truncated to test the binding ability to the PDGC21T aptamer, whereas to verify the second model, X-ray crystal structure of the CD49c-PDGC21T complex should be resolved [54]. The results of the competition assay implies that PDGC21T may inhibit CD49c and laminin interactions to control cancer cell growth, which will be further investigated.

Because CD49c is internalized after binding, it is a promising target for anti-tumor agent delivery [55,56]. Targeting CD49c for chemotherapy and gene therapy has two benefits for cancer control: blocking CD49c to prevent cancer cell metastasis and growth and delivering anti-tumor agents to induce cancer cell death.

## 5. Conclusions

Here, we identify a TNBC-bound aptamer that may act as a targeting ligand. Furthermore, we identified CD49c as a potential target for antibody or aptamer-guided therapy. Future in vitro and in vivo studies evaluating the therapeutic efficacy of PDGC21T-guided and CD49c-targeted therapies are warranted.

## Figures and Tables

**Figure 1 cancers-14-01570-f001:**
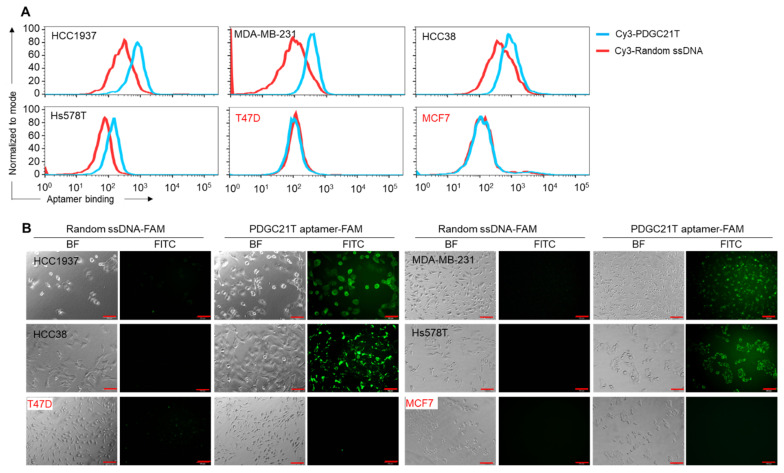
Specific binding of the PDGC21T aptamer to cultured triple-negative breast cancer (TNBC) cells. (**A**) Flow cytometry analysis demonstrates that PDGC21T aptamer binds to suspended TNBC cells but does not react with non-TNBC cells. (**B**) Fluorescent microscopy confirmed that PDGC21T binds to adherent TNBC cells but does not react with non-TNBC cells. Red font indicates non-TNBC cell line. BF; bright field, FITC; fluorescein isothiocyanate. Scale bars = 100 μm. The final incubation concentration of aptamers or random ssDNA was 200 nM.

**Figure 2 cancers-14-01570-f002:**
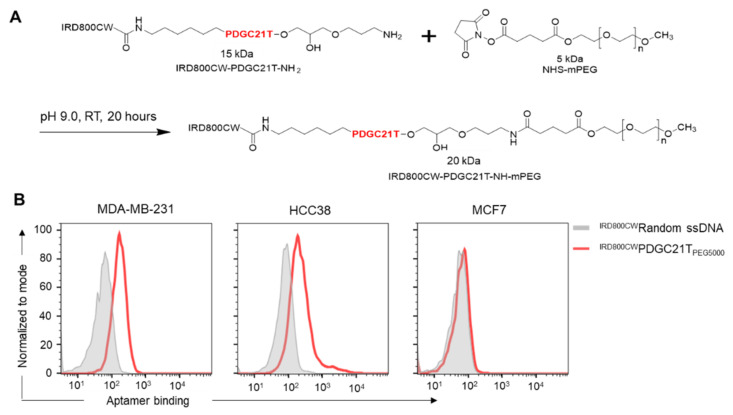
Preparation and validation of ^IRD800CW^PDGC21T_PEG5000_ aptamers. (**A**) Schematic showing ^IRD800CW^PDGC21T_PEG5000_ aptamer production. (**B**) Flow cytometry analysis of ^IRD800CW^PDGC21T_PEG5000_ aptamer binding to TNBC cells. The final incubation concentration of aptamer or random ssDNA was 200 nM.

**Figure 3 cancers-14-01570-f003:**
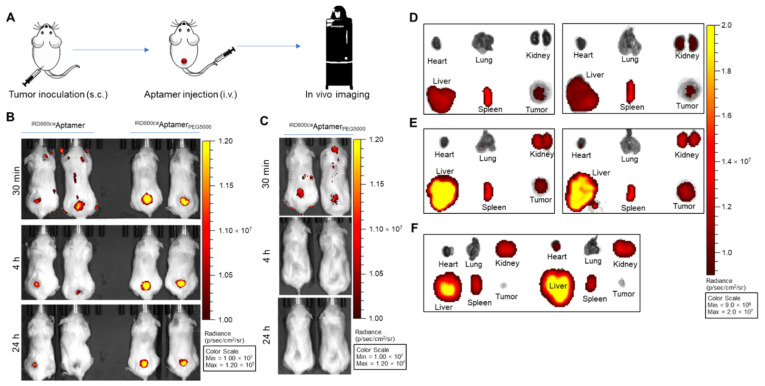
The ^IRD800CW^PDGC21T_PEG5000_ aptamer specifically targets TNBC xenograft tumors and has an extended in vivo half-life. (**A**) Flow diagram of the in vivo tumor targeting study using aptamer probes. The red sphere indicates the xenograft tumor site. s.c., subcutaneous; i.v., intravenous. (**B**) Relative to ^IRD800CW^PDGC21T aptamers, the signal enhancement from ^IRD800CW^PDGC21T_PEG5000_ aptamers was stronger and persisted longer in MDA-MB-231 tumors (up to 24 h post-aptamer administration). (**C**) In contrast, weak signals were observed peripherally in MCF7 tumor sites at 30 min post ^IRD800CW^PDGC21T_PEG5000_ aptamer administration but faded rapidly and became undetectable under the same imaging conditions. (**D**,**E**) Ex vivo imaging of resected MDA-MB-231 xenograft tumors, hearts, lungs, kidneys, livers, and spleens from mice treated with ^IRD800CW^PDGC21T aptamers (**D**), or ^IRD800CW^PDGC21T_PEG5000_ aptamers (**E**) post-whole-body imaging. (**F**) Ex vivo imaging of resected MCF7 xenograft tumors and major organs from mice treated with the ^IRD800CW^PDGC21T_PEG5000_ aptamers.

**Figure 4 cancers-14-01570-f004:**
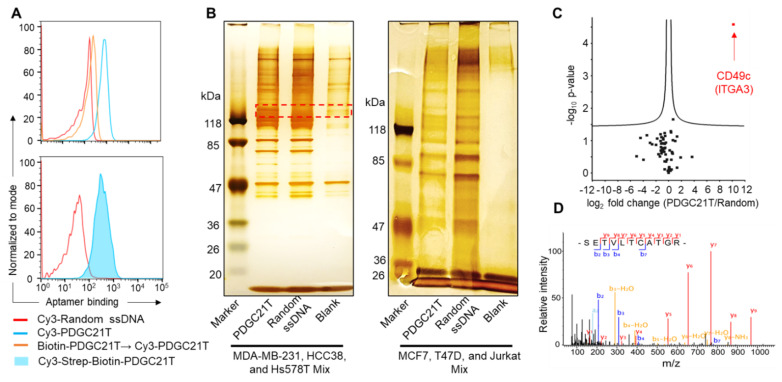
Identification of the PDGC21T aptamer target molecule. (**A**) Biotinylated PDGC21T aptamer targeted MDA-MB-231 cells with a similar binding capacity to that of Cy3-labeled PGC21T aptamers. In contrast, random ssDNA controls did not bind MDA-MB-231 cells. Biotin-PDGC21T→Cy3-PDGC21T indicates MDA-MB-231 cells that were first incubated with biotinylated PDGC21T and then incubated with Cy3-labeled PDGC21T. The final incubation concentration of aptamers or random ssDNA was 200 nM. (**B**) Aptamer-mediated immunoprecipitation assays. Membrane proteins derived from TNBC cells (MDA-MB-231, HCC38, and Hs578T mixtures) and non-TNBC cells (MCF7, T47D, and Jurkat mixture) were prepared and used for co-precipitation with biotinylated PDGC21T aptamer, biotinylated random ssDNA, or vehicle alone as a blank control. Resultant aptamer and target complexes were pulled down by streptavidin-immobilized agarose beads and separated on sodium dodecyl sulphate-polyacrylamide gel electrophoresis (SDS-PAGE), followed by silver staining. The red-dotted box shows the protein band of interest, which was collected for liquid chromatography tandem mass spectrometry (LC-MS/MS) identity analysis. The uncropped figures are shown in Appendix A. (**C**,**D**) LC-MS/MS identification of the target of PDGC21T. The proteins detected in PDGC21T aptamer and random sequence pull-down experiments were identified and quantified with label-free LC-MS. (**C**) Volcano plot of identified proteins. Each dot represents one protein. Proteins above the cutoff curves are statistically significant (Student’s *t*-test with 1% permutation-based FDR below 0.01). (**D**) Annotated MS/MS spectra of CD49c (ITGA3) peptide, STEVLTCATGR.

**Figure 5 cancers-14-01570-f005:**
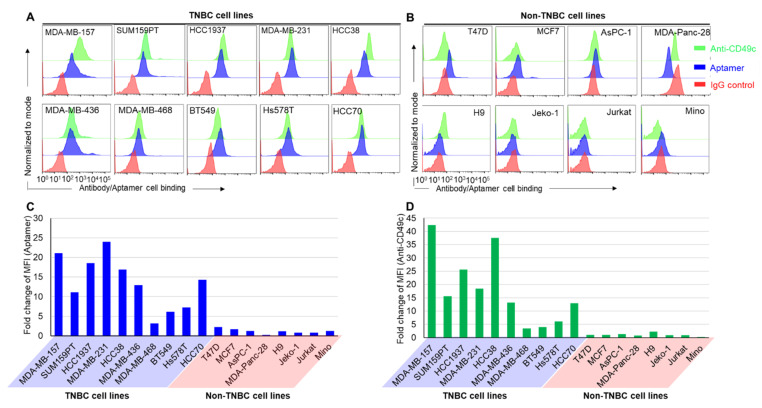
PDGC21T aptamer selectively targets CD49c-expressing TNBC cells. (**A**) Both PDGC21T aptamer and anti-CD49c targeted TNBC cells with high binding capacity but had (**B**) little or no binding to non-TNBC cells. Resultant mean fluorescent intensities of cell binding by aptamer (**C**) and antibody (**D**) were quantified by flow cytometry and graphed for comparison. The final incubation concentration of aptamer was 200 nM.

**Figure 6 cancers-14-01570-f006:**
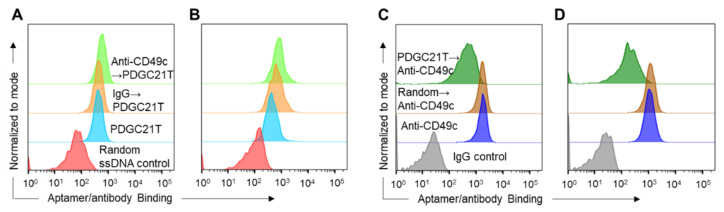
PDGC21T aptamer and anti-CD49c share an overlapping binding site on CD49c. Pre-treatment with anti-CD49c did not affect FAM-labeled PDGC21T aptamer binding to MDA-MB-231 (**A**) or HCC38 cells (**B**). IgG isotype and a random ssDNA were used as controls. Pre-treatment with PDGC21T aptamer inhibited PE-labeled anti-CD49c binding to MDA-MB-231 (**C**) and HCC38 cells (**D**). Arrows indicate incubation sequence of antibodies and aptamers.

## Data Availability

Data are contained within this article and in the Appendix A.

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
