# Peer review of "Aptamer Targets Triple-Negative Breast Cancer through Specific Binding to Surface CD49c"

_cancers, 2022, doi:10.3390/cancers14061570_

Round 1

Reviewer 1 Report

In reviewer opinion, this paper is enough to be approved, but there are two questions, so it was decided as a minor revision for the time being.

Two types of buffers were used in lines 92 and 94 on page 2, but reviewer thinks that washing with only binding buffer would have been sufficient. Why did you use different types? Rather, it seems that the composition of the binding buffer is more effective for washing.

In the experiment of Figure 2 on page 6, PEGylation was performed for a specific aptamer, but why was PEGylation not performed for the random sequence as a control?

Reviewer doesn't think additional experiments are necessary about the above, but would like you to answer these questions.

Author Response

In reviewer opinion, this paper is enough to be approved, but there are two questions, so it was decided as a minor revision for the time being.

RE: We appreciate the reviewer’s encouraging comments.

 Two types of buffers were used in lines 92 and 94 on page 2, but reviewer thinks that washing with only binding buffer would have been sufficient. Why did you use different types? Rather, it seems that the composition of the binding buffer is more effective for washing.

RE: Yes, we agree that the binding buffer can be used for washing.

Notably, the binding buffer is made of washing buffer with BSA and tRNA for blocking purposes. Cells have non-specific binding sites that require blocking prior to incubation with a specific targeting agent. That is the case when cells are incubated with antibodies, BSA or serum should be used to reduce the non-specific binding. For negatively charged DNA/RNA aptamers, this step also involves incubation with tRNA or salmon sperm (DNA) to block nucleic acid binding sites. In our methods, tRNA was used. Besides, DNA/RNA aptamers may attach non-specifically to positively charged proteins on cell surfaces due to their negative charge. We therefore also use BSA in our blocking and binding buffers as it has an isoelectric point of 4.7–5.2 and is hence negatively charged in a pH-neutral fluid.

Whereas in the washing process, there is unnecessary to supplement BSA and tRNA for blocking purposes, though they do not affect the washing efficacy. Since the usage of washing buffer was much more than that of binding buffer and considering the cost of experiments, we did not add BSA or tRNA in the washing buffer.

In the experiment of Figure 2 on page 6, PEGylation was performed for a specific aptamer, but why was PEGylation not performed for the random sequence as a control?

RE: Thank you for your comments. The PEGylated aptamer was synthesized for in vivo study because it can have a longer circulating lifetime for targeted therapy in the mouse model. Figures 2 and 6 are in vitro cell binding PEGylated aptamer, and random ssDNA sequence was used as assay background control.

In addition, it is difficult to produce a PEGylated random ssDNA library for several reasons. In solid-phase synthesis, labeling one terminal of the aptamer is easy, because the reaction can be carried out on solid phase. However, to label the other terminal of the aptamer, the reaction must be carried out in a solution phase. For a specific aptamer sequence, we can monitor the labeling reaction and purify aptamer products with both terminal modification by HPLC. In contrast, because the random aptamer library contains mixed sequences with different physical and chemical properties, it is difficult to monitor labeling and PEGylation reactions. Moreover, even the reaction works well, the separation and purification of PEGylated random sequences could be very difficult. We once tried to find a company to prepare the coupling product, but no company available provides such service.

Importantly, because PEGylation has no effect on the cell binding capacity of random ssDNA sequences, they are suitable for background signal control in cell-binding studies with PEGylated aptamers in vitro.

Reviewer 2 Report

The manuscript by Quanyuan et al describes PDGC21T aptamer that bound to TNBC cells and identified CD49c as a specific surface marker for TNBC cells using the aptamer-facilitated biomarker discovery technology. This manuscript is potentially interesting, but there are some points that need to be improved before publication.

Major comments:

Figure 3: Tumor accumulation is expected to vary depending on the cell line used to generate the Xenograft tumor (Figure 3B, S2, 5C). Usually, in such cases, cDNA or random ssDNA is used as a negative control to clarify that it is the effect of the aptamer. The authors should show such data.

Flow cytometry: The authors should provide information on the fluorescence intensity of the cells only as supporting information.

Figure 5: If aptamers and antibodies recognize the same epitope, there should be a correlation between the fold change of MFI of them, but the fold change of MFI of MDA-MB-231 and HCC38 are different. What is the cause of this? The expression level of CD49c in each cell line should be quantified by western blotting or other methods.

Line 320-322, Figure 6: “Different results from competition binding assays could be explained by possible cell-binding mechanisms of anti-CD49c and PDGC21T, with a potential interruption mechanism illustrated in Figures 6E and 6F.”, is anti-CD49c used a bispecific antibody that recognizes different epitopes of CD49c?

Line 358, Figure 2: “PEGylation did not affect PDGC21T binding.”, Kd values are important in evaluating aptamer and antibodies. The Kd values before and after PEG modification should be calculated with reference to previous studies.

Other comments:

Line 73: Please add information about the cell registration number.

Line 74-83: “(Corning)” or “(Corning, Corning, NY)”?

Line 92: Why did the authors use tRNA instead of DNA as a masking reagent for the DNA aptamer?

Line 103, 107: Please add information about anti-CD49c concentration.

Line 205-209: Please add purity and mass spectrometry information about IRD800CWPDGC21TPEG5000.

Line 260-261: Why did the authors use MDA-MB-231, HCC38, and Hs578T mixtures?

Figure 5: Why did the authors use IgG control as the negative control instead of random ssDNA as in other experiments?

Figure S1: What is the lower band of lane 2 (IRD800CWPDGC21T) of SDS-PAGE?

Reviewer 3 Report

In this manuscript, the authors described an interesting study that tested the specific binding of a previously reported aptamer to several TNBC cell lines and tumors in vivo. The molecular target of the aptamer was identified to be CD49c by co-precipitation and mass spectrometry. The study is interesting and important for the field of biomarker discovery. I suggest the acceptance of this manuscript for publication after minor revisions. Below are my specific suggestions.

  1. The introduction is quite short. I suggest that the authors expand the description of the utility of aptamers for the development of targeted therapy, since the authors have mentioned this application for several times in the manuscript (e.g. in the abstract). Although it is self-evident that aptamers can aid in the identification of molecular targets, it is not clear whether there is any advantage to use aptamers as therapeutic agents (compared to antibodies) once the molecular target is identified. For readers who are not familiar with aptamers, this is an important point.
  2. The authors need to explain better how the aptamer binding was determined by flow cytometry. The description of the method is short. Were the cells sorted according to whether the aptamers bind? If yes, why are there signals for control ssDNA? (Figure 1A). Do these signals come from non-specific binding? The X-axis of plots in Figure 1A is labeled as “aptamer binding”. What does “binding” really mean? Does the amount of right-shift of peaks correspond to binding affinity (or the expression level of target protein on cells)? The labels of scale bars in Figure 1B are not too small and not legible.
  3. The binding model presented in Figure 6E&F, which is based on the results of competition assays is not the only possible model. The data does not exclude the possibility that the aptamer is an allosteric inhibitor of antibody binding. The fact that the binding of the aptamer reduces the binding of antibody does not mean that these two molecules share binding sites. Inhibition can be achieved through allosteric interactions. The fact that the aptamer can still bind when the antibody is already bound to the receptor seems to be more consistent with allosteric inhibition model, though the possibility that the aptamers can bind to multiple sites cannot be eliminated. If the authors believe that the model presented in Figure 6E&F is the only possible model, they should provide a rationale for such a claim.

Reviewer 4 Report

In this study the authors investigated the affinity of PDGC21T aptamer for poor differentiated TNBC tumor cells. They showed the affinity of the PDGC21T aptamer against TNBC both in in vitro and in cell xenografts in mice. Furthermore, they identified one of the binding sites on the cell surface protein CD49c (integrin α3). These results offer PDGC21T aptamer as a promising therapeutic agent to target TNBC cells.

Minor edits:

Line 137 please use the word vertically, with "e"

Figure 6C has the word Anti misspelled as Anit, please revise.

Author Response

In this study the authors investigated the affinity of PDGC21T aptamer for poor differentiated TNBC tumor cells. They showed the affinity of the PDGC21T aptamer against TNBC both in in vitro and in cell xenografts in mice. Furthermore, they identified one of the binding sites on the cell surface protein CD49c (integrin α3). These results offer PDGC21T aptamer as a promising therapeutic agent to target TNBC cells.

RE: Thank reviewer for your encouraging comments.

Minor edits:

Line 137 please use the word vertically, with "e"

RE: Thank reviewer again. Sorry for the confusion, we incubated aptamer and cells in a rotator, so it should be vortical incubation rather than vertical incubation. To avoid any misunderstanding, we removed the "vortically" and change the sentence as "...and tubes were incubated with rotation for 1 h at 4ËšC"

Figure 6C has the word Anti misspelled as Anit, please revise.

RE: We appreciate your scrupulous observation. The typo has been corrected in the revision.

Round 2

Reviewer 2 Report

Figure 3: The authors responded, “Our results indeed showed that the signals were different among different Xenograft tumors. We assumed that situation should be caused by the following reasons. First, the expression levels of CD49c in different xenograft tumors were different as we tested in vitro by using an anti-CD49c antibody (Figure 5). Second, the abundance of vessels around different xenograft tumors may be different, leading to the different abundance of aptamer around xenograft tumors.” That is, it is not enough evidence to support the authors' assertion, because it cannot be determined from the current experimental results alone whether the main factor of tumor accumulation is CD49c recognition by aptamers or just EPR effects. The authors should carry out comparative studies with negative control nucleotides against representative cell lines. As the authors say, it is difficult to purify PEGylated random sequences by HPLC, but it is easier to purify by PAGE. In addition, a single sequence such as complementary strand or scrambled sequence of the aptamer is often used as negative control nucleotides, which can be purified in the same way as aptamers.

Round 3

Reviewer 2 Report

The manuscript has been much improved and is in nice condition now.